# Neural Lyapunov Control of Unknown Nonlinear Systems with Stability Guarantees

**Ruikun Zhou**
Department of Applied Mathematics
University of Waterloo
ruikun.zhou@uwaterloo.ca

**Thanin Quartz**
Department of Applied Mathematics
University of Waterloo
tquartz@uwaterloo.ca

**Hans De Sterck**
Department of Applied Mathematics
University of Waterloo
hans.desterck@uwaterloo.ca

**Jun Liu**
Department of Applied Mathematics
University of Waterloo
j.liu@uwaterloo.ca

## Abstract

Learning for control of dynamical systems with formal guarantees remains a challenging task. This paper proposes a learning framework to simultaneously stabilize an unknown nonlinear system with a neural controller and learn a neural Lyapunov function to certify a region of attraction (ROA) for the closed-loop system. The algorithmic structure consists of two neural networks and a satisfiability modulo theories (SMT) solver. The first neural network is responsible for learning the unknown dynamics. The second neural network aims to identify a valid Lyapunov function and a provably stabilizing nonlinear controller. The SMT solver then verifies that the candidate Lyapunov function indeed satisfies the Lyapunov conditions. We provide theoretical guarantees of the proposed learning framework in terms of the closed-loop stability for the unknown nonlinear system. We illustrate the effectiveness of the approach with a set of numerical experiments.

## 1 Introduction

Finding the region of attraction (ROA) of an asymptotically stable equilibrium point for a nonlinear dynamical system remains one of the most challenging tasks, especially for systems with uncertainty [26]. Even though the well-known Lyapunov theorems which use the level sets of a Lyapunov function to estimate the ROA were proposed more than a century ago, there is no general approach to finding a valid Lyapunov function for general nonlinear systems with provable guarantees [22]. Thanks to the development of learning-based methods, several data-driven approaches have recently shown their effectiveness in identifying unknown (or partly known) systems [8, 30]. However, how to simultaneously find Lyapunov functions and nonlinear controllers for systems with unknown dynamics is still an open and active problem in control and robotics applications [20].

By exploiting the expressiveness of neural networks to tackle the complex nonlinearities of dynamical systems, we propose a learning framework for simultaneously learning control functions and neural Lyapunov functions. We build upon the framework in [5], but address the more challenging problem of stabilizing a nonlinear system with unknown dynamics with formal stability guarantees. In addition, we complement the computational framework in [5], by proving the existence of a neural network satisfying the Lyapunov conditions, except on an arbitrarily small neighborhood of the origin, which offers a theoretical basis for the learning framework here and in [5].

36th Conference on Neural Information Processing Systems (NeurIPS 2022).

Our proposed learning framework consists of two neural networks. The first neural network learns the unknown dynamics, while the second one aims to identify a valid Lyapunov function and a stabilizing nonlinear controller. After training the neural networks, we derive error bounds that can be verified directly by the satisfiability modulo theories (SMT) solver. We then use Lipschitz continuity of the nonlinear dynamics and its neural approximation to prove rigorous theoretical guarantees on the closed-loop stability of the unknown system under the learned nonlinear controller. To the best knowledge of the authors, no prior work has provided closed-loop stability analysis in such a general setting of simultaneously learning the dynamics, a nonlinear controller, and a Lyapunov function.

We experimented on three typical nonlinear system examples: the Van der Pol oscillator, the inverted pendulum, and the unicycle path following. To align with the theoretical results, we carefully establish bounds for the Lipschitz constants, keep track of the training losses, and adjust the Lyapunov conditions to be verified by the SMT solver accordingly so that the verified ROA is valid for the unknown system. The numerical experiments show that the ROA learned this way are comparable with the ones computed with actual dynamics, and improve that obtained from the LQR controller.

**Related work.** Artificial neural networks are capable of estimating any nonlinear function with arbitrarily desired accuracy [18]. As both the right-hand side of a differential equation and the Lyapunov function can be regarded as nonlinear (or linear) functions, mapping from the state spaces to some vector spaces, a natural question is: can we use neural networks to approximate the dynamics and find a Lyapunov function to attain a larger estimate of the ROA, compared with widely used linear–quadratic regulator (LQR) and sum of squares (SOS) methods [29, 23]?

Various applications of neural networks for system identification and Lyapunov stability have been proposed recently [20, 15]. For instance, Wang et al. [19] use a recurrent neural network to generate the state-space representation of an unknown dynamical system. A Lyapunov neural network is derived in [32] to learn a fixed quadratic format Lyapunov function for finding the largest estimated ROA. However, when it comes to finding a Lyapunov function with a typical feedforward neural network to certify the stability for the nonlinear systems, performing this kind of regression task is difficult for shallow neural networks, where only equality and/or inequality Lyapunov conditions, instead of explicit expression or values of the function, are given. Consequently, it is crucial to develop methods to test or verify the outputs of a neural network. There are two main approaches to tackle this issue. One is converting the neural networks into a mixed-integer programming (MIP) format which can be solved by MIP solvers. The other is to encode the verification problem into an SMT problem. In the first category, [7, 6] find a robust Lyapunov function for an uncertain nonlinear system or a hybrid system which can be partly approximated by piecewise linear neural networks with the help of a counterexample-guided method using mixed integer quadratic programming (MIQP), and the corresponding robust ROA is estimated. With a similar technique, [10] synthesizes a neural-network controller and Lyapunov function simultaneously to certify the dynamical system's stability by converting the dynamics, the Lyapunov function, and the controller into piecewise linear functions and solving it with mixed-integer linear programming (MILP). On the other hand, several SMT solvers have been developed by various scholars [28, 21], for instance, dReal [17], with which tasks with various nonlinear real functions can be elegantly handled. This can be employed in most cases for verifying existing neural network structures, e.g., with $\mathrm{tanh}$ and $\mathrm{sigmoid}$ activation functions. With the help of these SMT solvers, Chang et al. propose an algorithm to find and validate a neural Lyapunov function satisfying the Lyapunov conditions with a one-hidden-layer neural network [5]. More recent work in [11] uses the same methodology to approximate control Lyapunov barrier functions for nonlinear systems to guarantee both stability and safety for reach-avoid robot tasks, while [36] uses a similar framework to design stabilizing controllers for unknown nonlinear systems with an implementation of Koopman operator theory. Gaussian process (GP) regression is used in [1, 2] to learn the unknown systems while providing safety and stability guarantees for the states in the ROA with high probability, at the same time how to enlarge the estimated ROA is also well studied. On top of that, a method to obtain a larger ROA by iteratively re-designing a control Lyapunov function is presented in [27]. Apart from the above traditional machine learning methodologies, [12] proposes a piecewise learning framework with piecewise affine models, where stability guarantees are verified with quadratic Lyapunov functions by solving an optimization problem. However, we have to contend with the safety-critical nature of certain control and robotics applications. As a result, worse-case bounds and performance guarantees are more desirable. In a similar vein, an upper bound on the infinity-norm bounds were established in the context of training deep residual networks [25]. We remark that none of these works analyzed the closed-loop stability in the general setting as ours.

**Contributions.** We summarize the main contributions of the paper as follows.

- We propose a framework for simultaneously learning a nonlinear system, a stabilizing nonlinear controller, and a Lyapunov function for verifying the closed-loop stability of the unknown system.

- We provide theoretical analysis on the existence of a neural Lyapunov function that verifies the Lyapunov conditions, except on an arbitrarily small neighborhood of the origin, provided that the origin is asymptotically stable. This provides theoretical support for the proposed method as well as other neural Lyapunov frameworks in the literature.

- Combining classic Lyapunov analysis with rigorous error estimates for neural networks, we establish closed-loop stability for the unknown system under the learned controller. This is accomplished with the help of SMT solvers and recent tools for estimating Lipschitz constants for neural networks.

## 2 Preliminaries

Throughout this work, we consider a nonlinear control system of the form

$$\dot{x} = f(x, u), \quad x(0) = x_0, \tag{1}$$

where $x \in \mathcal{D}$ is the state of the system, and $\mathcal{D} \subseteq \mathbb{R}^n$ is an open set containing the origin; $u \in \mathcal{U} \subseteq \mathbb{R}^m$ is the feedback control input given by $u = \kappa(x)$, where $\kappa(x)$ is a continuous function of $x$. Without loss of generality, we assume the origin is an equilibrium point of the closed-loop system

$$\dot{x} = f(x, \kappa(x)), \quad x(0) = x_0. \tag{2}$$

When there is no ambiguity, we also refer to the right-hand side of (2) by $f$.

We assume that we do not have explicit knowledge of the right-hand side of the system (1). The main objective is to stabilize the unknown dynamical system by designing a feedback control function $\kappa$. Stability guarantees are established using Lyapunov functions. We next present some preliminaries on model assumptions and Lyapunov stability analysis.

**Model assumptions**

**Assumption 1** (Lipschitz Continuity). *The right-hand side of the nonlinear system (1) is assumed to be Lipschitz continuous, i.e.,*

$$\|f(x, u) - f(y, v)\| \le L\|(x, u) - (y, v)\| \quad \forall x, y \in \mathcal{D} \quad and \quad \forall u, v \in \mathcal{U},$$

*where $L$ is called the Lipschitz constant; $(x, u)$ and $(y, v)$ denote the concatenation of the corresponding two vectors. We assume the Lipschitz constant $L$ is known.*

**Assumption 2** (Partly Known Dynamics). *The linearized model about the origin in the form of $\dot{x} = Ax + Bu$, where $A$ and $B$ are constant matrices, is known for the nonlinear system (1).*

To analyze the stability properties of the closed-loop system for (1) in the presence of uncertainty (due to the need to approximate the unknown dynamics), we introduce a more general notion of stability about a closed set $A$ (see, e.g., [24], and the Appendix for a definition). When $A = \{0\}$, this coincides with the standard notion of stability about an equilibrium point. Intuitively, set stability w.r.t. to $A$ is measured by closeness and convergence of solutions to the set $A$. To this end, define the distance from $x$ to $A$ by $\|x\|_A = \inf_{y \in A} \|x - y\|$.

Set invariance [3] plays an important role in our analysis.

**Definition 1** (Forward Invariance). *A set $\Omega \subset \mathbb{R}^n$ is said to be forward invariant for (2) if $x_0 \in \Omega$ implies that $x(t) \in \Omega$ for all $t \ge 0$.*

**Definition 2** (Region of Attraction). *For a closed forward invariant set $A$ that is uniformly asymptotically stable (UAS), the region of attraction is the set of initial conditions in $\mathcal{D}$ such that the solution for the closed-loop system (2) is defined for all $t \ge 0$ and $\|x(t)\|_A \to 0$ as $t \to \infty$.*

**Remark 1.** *Any set satisfying Definition 2 is called a region of attraction. The ROA is the largest set contained in $\mathcal{D}$ satisfying Definition 2.*

**Definition 3** (Lie Derivatives). *The Lie derivative of a continuously differentiable scalar function $V : \mathcal{D} \to \mathbb{R}$ over a vector field $f$ and a nonlinear controller $u$ is defined as*

$$\nabla_f V(x) = \sum_{i=1}^{n} \frac{\partial V}{\partial x_i} \dot{x}_i = \sum_{i=1}^{n} \frac{\partial V}{\partial x_i} f_i(x, u). \tag{3}$$

The lie derivative measures the rate of change along the system dynamics.

The following is a standard Lyapunov theorem for the UAS property of a compact invariant set.

**Theorem 1** (Sufficient Condition for UAS property). *Consider the closed-loop nonlinear system* (2). *Let $A \subset \mathcal{D}$ be a compact invariant set of this system. Suppose there exists a continuously differentiable function $V : \mathcal{D} \to \mathbb{R}$ that is positive definite with respect to A, i.e.,*

$$V(x) = 0 \ \forall x \in A \ \text{and} \ V(x) > 0 \ \forall x \in \mathcal{D} \setminus A, \tag{4}$$

*and the lie derivative is negative definite with respect to A, i.e.*

$$\nabla_f V(x) < 0 \ \forall x \in \mathcal{D} \setminus A. \tag{5}$$

*Then, A is UAS for the system.*

See [24, 35] for sufficiency and necessity of Lyapunov conditions for set stability under more general settings. The function $V$ satisfying (4) and (5) is called a Lyapunov function with respect to $A$.

From the forward invariance of level sets of a Lyapunov function it follows that these sets provide an estimate of the ROA (see [22]).

**Lemma 1** (Region of Attraction with Lyapunov Functions). *Suppose that $V$ satisfies the conditions in Theorem 1. Denote*

$$V^c := \{ x \in \mathcal{D} \mid V(x) \le c \}.$$

*For every $c > 0$, $V^c$ is a region of attraction for the closed-loop system* (2).

## 3 Learn and Stabilize Dynamics with Neural Lyapunov Functions

In this section, we show how to use two shallow neural networks to learn the dynamics and find a valid neural Lyapunov function. The first network is designed to learn the dynamics as shown in (1), and the second network aims to identify a valid neural Lyapunov function and its corresponding nonlinear controller. The algorithmic structure can be found in Fig. 1 and the pseudocode in Algorithm 1.

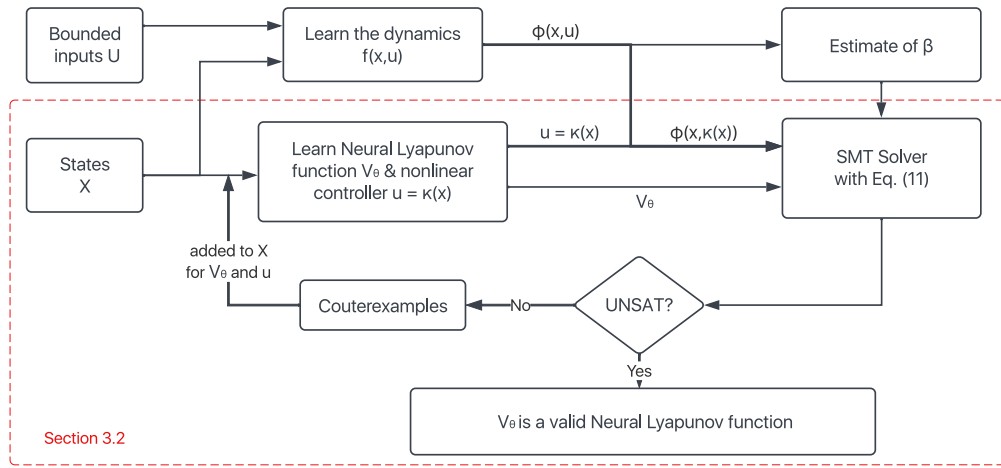

Figure 1: The schematic diagram of the proposed algorithm.

### 3.1 Learn the Unknown Dynamics

According to the well-known universal approximation theorem [9], shallow neural networks are able to approximate the right-hand side $f(x, u)$ of the nonlinear system (1) arbitrarily well. Here we use a one-hidden-layer feedforward neural network, denoted as $\phi(x, u)$, to conduct this regression task with a mean square loss. For notational convenience, we slightly overuse the notation of $\phi$ to denote both the the learned dynamics and the parameters that define the function. Note that if we use the $\mathtt{tanh}$ or $\mathtt{sigmoid}$ function as activation functions, the output layer should not have such activation functions, as $f$ is not bounded in most cases.

When learning the dynamics, $f$ is regarded as a function of two variables, $x$ and $u$. The input dimension of the neuron network is $(n + m)$, and output dimension is $n$. The training data of $x$ and $u$ are sampled uniformly and independently over their respective spaces. In most robotic systems the input is provided by motors, which is typically saturated in practice. To reflect this, we use the activation function $\tanh$ to simulate this property. The format of the controller is

$$u = \kappa(x) = C\sigma(kx + b), \tag{6}$$

where $C$ is a constant matrix, determined by the saturation property of the controller in real systems, which defines the bounds of $\mathcal{U}$. $k$ and $b$ are the weight matrix and bias vector, respectively, which are initialized with an LQR controller based on the linearized system $\dot{x} = Ax + Bu$. The bias $b$ is chosen such that $f(0, C\sigma(b)) = 0$, i.e., the origin is an equilibrium point for the closed-loop system (2). Note that the bias vector $b$ is not updated in the learning process.

## 3.2 Neural Lyapunov Function and Nonlinear Controller

Following learning the dynamics using $\phi$, the other neural network is designed to learn a nonlinear controller and a neural Lyapunov function simultaneously during the same training process. The detailed structure can be found in the Appendix. It is worth mentioning that since we train the Lyapunov function with data generated by the learned dynamics, which is an approximation of the actual dynamics, we rewrite (4) and (5) as follows to accommodate for approximation errors:

$$V(0) = 0, \text{ and }, \forall x \in \mathcal{D}\backslash\{0\}, V(x) > 0 \text{ and } \nabla_\phi V(x) < -\beta. \tag{7}$$

Here $\beta$ is a positive real number, which can be determined as follows.

Assume that $(x, u)$ is a pair of unsampled state and inputs, and $(y, v)$ is its nearest known sample used in training or testing the neural network for learning the dynamics. Let $\delta > 0$ be such that $\|(x, u) - (y, v)\| \le \delta$ holds for all such $(x, u)$ and $(y, v)$ and let $M > 0$ be such that $\|\frac{\partial V}{\partial x}\| < M$ for all $x \in \mathcal{D}$. Denote $\alpha$ as the maximum of the 2-norm loss among all known samples, which can be from either the training dataset or the test dataset. Then the bound on the generalization error can be calculated as

$$\|f(x, u) - \phi(x, u)\| \le \|f(x, u) - f(y, v)\| + \|f(y, v) - \phi(y, v)\| + \|\phi(y, v) - \phi(x, u)\|$$
$$\le K_f\delta + \alpha + K_\phi\delta < \frac{\beta}{M}, \tag{8}$$

where $f$ and $\phi$ are Lipschitz continuous with respective Lipschitz constants $K_f, K_\phi$ and $\beta > 0$ is some sufficiently large constant. Then, choosing $\beta$ to satisfy (8) implies that

$$\nabla_f V(x) - \nabla_\phi V(x) \le \|\frac{\partial V}{\partial x}\| \|f(x, \kappa(x)) - \phi(x, \kappa(x))\| < M\frac{\beta}{M} = \beta. \tag{9}$$

In view of (7) and (9), we have

$$\nabla_f V(x) < \nabla_\phi V(x) + \beta < -\beta + \beta = 0, \quad \forall x \in \mathcal{D}\backslash\{0\}, \tag{10}$$

which implies that the actual dynamics is also stable with the obtained neural Lyapunov function.

Here, we calculate $K_\phi$ by using LipSDP-network developed in [13], and $M$ can be determined by checking the inequality with an SMT solver. An initial guess of $\beta$ is needed according to some prior knowledge, and it will be re-computed in each epoch with (8).

A valid neural Lyapunov function can be guaranteed by the SMT solver with all the Lyapunov conditions written as falsification constraints in the form of first-order logic formula over reals [5]:

$$\Phi_\varepsilon(x) := \left(\sum_{i=1}^n x_i^2 \ge \varepsilon\right) \wedge \left(V(x) \le 0 \vee \nabla_\phi V(x) \ge -\beta\right), \tag{11}$$

where $\varepsilon$ is a numerical error parameter, which is explicitly introduced for controlling numerical sensitivity around the origin. If the SMT solver returns either `UNSAT` this means that the falsification constraint is guaranteed not to have any solutions and confirms all the Lyapunov conditions are met. If the SMT solver returns $\delta$-`SAT`, this means there exists at least one counterexample under the $\delta$-weakening condition [16] that satisfies the falsification conditions.

We use $\theta$ to denote the parameter vectors for a neural Lyapunov function candidate $V_\theta$. The parameters $\theta$ and $k$ are found by minimizing the following cost function, which is a modification of the so-called *empirical Lyapunov risk* in [5] by adding one more term $\|\frac{\partial V}{\partial x}\|$, as we need $\beta$ to be small as well:

$$L(\theta, k) = \frac{1}{N} \sum_{i=1}^{N} \left( C_1 \max\left(-V_\theta\left(x_i\right), 0\right) + C_2 \max\left(0, \nabla_\phi V_\theta\left(x_i\right)\right) \right) + C_3 V_\theta^2(0) + C_4 \|\frac{\partial V}{\partial x}\| \quad (12)$$

where $C_1$, $C_2$, $C_3$ and $C_4$ are tunable constants. The cost function can be regarded as the positive penalty of any violation of the Lyapunov conditions in (4) and (5). Note that the ROA can also be maximized by adding an $L_2$-like cost term to the *Lyapunov risk* with $L(\theta, k) + \frac{1}{N}\sum_{i=1}^{N}\|x_i\|_2 - \alpha V_\theta\left(x_i\right)$, where $\alpha$ is a tunable parameter, as shown in the original paper [5].

---

**Algorithm 1** Neural Lyapunov Control with Unknown Dynamics

---

1: **function** LEARNINGDYNAMICS($X_{dyn}, U_{(bdd)}$)
2:     Set learning rate ($\gamma$), input dimension ($n + m$), output dimension ($n$)
3:     **repeat**
4:         $f \leftarrow \text{NN}_\phi(x)$                                ▷ Output of forward pass of neural network
5:         Compute MSE $L(f, \phi)$
6:         $\phi \leftarrow \phi - \gamma\nabla_\phi L(f, \phi)$                       ▷ Updates Weights using SGD
7:     **until** convergence
8:     **return** $\phi$
9: **end function**
10: **function** LEARNINGLYAPUNOV($X_{Lyp}, f_\phi, k^{lqr}$)
11:     Set learning rate ($\alpha$), input dimension ($n$), output dimension (1)
12:     Initialize feedback controller u to LQR solution $k^{lqr}$
13:     **repeat**
14:         $V_\theta(x), u(x) \leftarrow NN_{\theta,u}(x)$                   ▷ Output of forward pass of neural network
15:         $\nabla_\phi V(x) \leftarrow \sum_{i=1}^{D_{in}} \frac{\partial V}{\partial x_i}[\phi]_i(x)$
16:         Compute Lyapunov risk $L(\theta, k)$
17:         $\theta \leftarrow \theta - \alpha\nabla_\theta L(\theta, k)$
18:         $k \leftarrow k - \alpha\nabla_k L(\theta, k)$                        ▷ Updates Weights using SGD
19:     **until** convergence
20:     **return** $V_\phi, u$
21: **end function**
22: **function** FALSIFICATION($f_\phi, u, V_\theta, \varepsilon, \delta^*, \beta$)
23:     Encode conditions from (11)
24:     Use SMT solver with $\delta$ to verify the conditions
25:     **return** satisfiability
26: **end function**
27: **function** MAIN( )
28:     **input** initial guess of bound ($\beta$), parameters of LQR ($k^{lqr}$), radius ($\varepsilon$), precision ($\delta^*$), sampled states $X$, sampled inputs $U$
29:     $\phi \leftarrow$ LEARNINGDYNAMICS($X_{dyn}, U_{(bdd)}$)
30:     **while** Satisfiable **do**
31:         Add counterexamples to $X$
32:         $V_\phi, u \leftarrow$ LEARNING-LYAPUNOV($X_{Lyp}, \phi, k^{lqr}$)
33:         update $\beta$ according to (8)
34:         CE $\leftarrow$ FALSIFICATION($\phi, u, V_\theta, \varepsilon, \delta^*, \beta$)
35:     **end while**
36: **end function**

---

## 4   Theoretical Guarantees

In this section, we analyze the theoretical guarantees of using neural networks to learn to control an unknown nonlinear system and a Lyapunov function for certifying the closed-loop stability.

## 4.1 Approximation guarantee of unknown dynamics

Our analysis provides stability guarantees for the unknown system by rigorously quantifying the errors using Lipschitz constants of the unknown dynamics and its neural approximation. To this end, we need a theoretical guarantee that extends the universal approximation theorem, stating that we can approximate the Lipschitz constants and function values by a neural network to an arbitrary precision.

**Theorem 2.** *(Approximation of Lipschitz constants). Suppose that $K \subset \mathbb{R}^n$ is a compact set.*
*(a) If $f : K \to \mathbb{R}^m$ is L-Lipschitz in the uniform norm, i.e.*

$$\|f(x) - f(y)\|_\infty \le L\|x - y\|_\infty, \tag{13}$$

*then for every $\epsilon > 0$ there exists a neural network of the form $\phi(x) = C(\sigma \circ (\omega x + b))$ for $\sigma \in C^1(\mathbb{R})$ and not a polynomial, $\omega \in \mathbb{R}^{k \times m}, b \in \mathbb{R}^k$ and $C \in \mathbb{R}^{k \times n}$ for some $k \in \mathbb{N}$ such that $\sup_{x \in K} |f(x) - \phi(x)| < \epsilon$ and $\phi$ has a Lipschitz constant of $L + \epsilon$ in the same norms as ( 13).*
*(b) If $f : K \to \mathbb{R}^m$ is L-Lipschitz in the two norm, i.e.*

$$\|f(x) - f(y)\|_\infty \le L\|x - y\|_2, \tag{14}$$

*then for every $\epsilon > 0$ there exists a neural network $\phi$ of the same form such that $\sup_{x \in K} |f(x) - \phi(x)| < \epsilon$ and $\phi$ has a Lipschitz constant of $L + \epsilon \left( \frac{\sqrt{n+n/\epsilon}}{2} + L \right)$ in the same norms as (14).*

The idea of the proof is to first approximate $f$ by a smooth function $F$ and then approximate $F$ by a neural network $\phi$ and the details can be found in the Appendix.

**Remark 2.** *The equivalence of norms gives an upper bound on the Lipschitz constant for all norms.*

## 4.2 Existence of neural Lyapunov functions

As a theoretical guarantee we show that it is possible to train a neural network as a Lyapunov function, provided that a Lyapunov function exists. According to converse Lyapunov theorems [24, 35], Lyapunov functions do exist when the origin is UAS. More specifically, we show that the learned neural network satisfies the Lyapunov conditions outside some neighborhood of the origin that can be chosen to be arbitrarily small in measure. The idea will be to perform an under-approximation of the domain $\mathcal{D}$ in a controlled way. Let $(\mathbb{R}^n, \mathcal{B}(\mathbb{R}^n), \mu)$ denote the standard measure space where $\mathcal{B}(\mathbb{R}^n)$ is the Borel $\sigma$-algebra and $\mu$ is the Lebesgue measure. The following lemma from [33] states that it is possible to arbitrarily under-approximate open sets by compact sets in measure.

**Lemma 2.** *For every open set $O \in \mathcal{B}(\mathbb{R}^n)$ such that $\mu(O) < \infty$ and every $\epsilon > 0$, there exists a compact set $K$ such that $\mu(O \setminus K) < \epsilon$.*

By the pointwise approximation of the universal approximation theorem, it is not possible to satisfy the Lyapunov conditions on $\mathcal{D}$ as this set contains the origin. However, if there is a Lyapunov function for (2) that a neural network could learn and if practical stability (i.e., set stability w.r.t. a sufficiently small neighborhood of the origin) is sufficient, Theorem 3 states that there exists a neural network satisfying the Lyapunov conditions on a compact set $K$ except on a closed neighborhood $B$ of the origin that is UAS. Moreover, this approximation can be controlled in measure. The details of the proof can be found in the Appendix.

**Theorem 3.** *Suppose that the origin is UAS for system (2) and $\mathcal{I}$ is a forward invariant set contained in the ROA of the origin. Fix any $\gamma_1, \gamma_2 > 0$. There exists a forward invariant and compact set $K \subset \mathcal{I}$ satisfying the under approximation $\mu(\mathcal{I} \setminus K) < \gamma_1$. On $K$ there exists a neural network $V_\phi$ that satisfies the Lyapunov conditions on $K \setminus \mathcal{A}$, where $\mathcal{A}$ is a closed neighborhood of the origin. The neural Lyapunov function $V_\phi$ can certify that a closed invariant set $\mathcal{B}$ containing $\mathcal{A}$ and satisfying $\mu(\mathcal{B} \setminus \mathcal{A}) < \gamma_2$ is UAS. Furthermore, the set $K$ is contained in the ROA of $\mathcal{B}$.*

An important feature of Lyapunov functions is that the level sets approach the ROA. We can show that the neural Lyapunov function $V_\phi$ has such a property. Details can be found in the Appendix.

## 4.3 Asymptotic Stability Guarantees of Unknown Nonlinear Systems

With the theoretical guarantee of learning a neural Lyapunov function established above, we show that the neural network trained with the learned dynamics is robust, that is this neural network also

satisfies the Lyapunov conditions with respect to the actual dynamics $f$. Since the SMT solver verifies the Lyapunov conditions outside of some $\epsilon$-ball which is not necessarily forward invariant, the following technical assumption helps bridge this gap. The assumption is mild, because for the nonlinear system to be stabilizable, it is reasonable to assume that it has a stabilizable linearization. A linear system $\dot{x} = Ax + Bu$ is said to be *stabilizable* if there exists a matrix $\mathcal{K}$ such that $A + B\mathcal{K}$ is Hurwitz, i.e., all eigenvalues of $A + B\mathcal{K}$ have negative real part. If $(A, B)$ is stabilizable, then the gain matrix $\mathcal{K}$ can be obtained by an LQR controller.

**Assumption 3** (ROA of LQR Controller). *Suppose that the linearized model $\dot{x} = Ax + Bu$ from Assumption 2 is stabilizable. Consequently, an LQR controller and a quadratic Lyapunov function can be found such that the origin is UAS for the closed-loop system* (2). *Furthermore, we assume that the set $B_\varepsilon$ which is not verified by a SMT solver lies in the interior of a ROA of the closed-loop system, provided by the quadratic Lyapunov function.*

Since this $\varepsilon$-ball is small, we further assume that the level sets of the Lyapunov function are contained in the ROA of the LQR controller.

**Assumption 4** (Controlled Level Sets). *Denote $B_\varepsilon := \{x : \|x\| \leq \varepsilon\}$. Let $V$ be a continuously differentiable function satisfying the Lyapunov conditions on $\mathcal{D} \setminus B_\varepsilon$. Suppose that there exists constants $0 < c_1 < c_2$ such that the following chain of inclusions holds*

$$\{x \in \mathcal{D} : V(x) \leq c_1\} \subset B_\varepsilon \subset \{x \in \mathcal{D} : V(x) \leq c_2\}, \tag{15}$$

*and $\{x \in \mathcal{D} : V(x) \leq c_2\}$ lies in the interior of the ROA of the closed loop system provided by the quadratic Lyapunov function.*

**Theorem 4.** *(Stability Guarantees for the Unknown System) Let $\phi$ be the approximated dynamics of right-hand side of the closed-loop system* (2) *trained by the first neural network. There exists a neural Lyapunov function $V$ which is learned using $\phi$ and verified by an SMT solver that satisfies the Lyapunov conditions with respect to the actual dynamics $f$. Furthermore, if the system satisfies Assumption 3 and $V$ satisfies Assumption 4, then the origin is UAS for the closed-loop system* (2).

The details of this proof can be found in the Appendix. This theorem proves that if the dynamics are approximated to sufficient precision then the neural Lyapunov function satisfies the Lyapunov conditions on $\mathcal{D} \setminus B_\varepsilon$ for the actual dynamics. Furthermore, if the level sets of the neural Lyapunov function are sufficiently well behaved and the set $B_\varepsilon$ excluded from SMT verification is small then this learning framework certifies that the origin is UAS for the actual system (2).

## 5 Experiments

In this section, we demonstrate the effectiveness of the proposed algorithm on learning the unknown dynamics and finding provably stable controllers as well as neural Lyapunov functions for various classic nonlinear systems. In all the following problems, we use a two-layer neural network with one hidden layer for both learning the dynamics and learning the neural Lyapunov function. For learning the dynamics, the number of neurons in the hidden layer varies from 100 to 200 without an output layer activation function as stated before, and we call this neural network FNN for convenience. However, for learning the neural Lyapunov function, there are six neurons in the hidden layer for all the experiments, and we name this neural network VNN in short. Regarding other parameters, we use the Adam optimizer for both FNN and VNN, and we use dReal as the SMT solver, setting the precision $\delta^*$ for the falsification as 0.01 for all experiments. All the training of FNN is performed on Google Colab with a 16GB GPU, and VNN training is done on a 3 GHz 6-Core Intel Core i5.

### 5.1 Van der Pol Oscillator

As a starting point, we test the proposed algorithm on the nonlinear system without input $u$ first to show its effectiveness in learning unknown dynamical systems and finding the valid neural Lyapunov function. The Van der Pol oscillator is a well-known nonlinear system with stable oscillations as a limit cycle. The area within the limit cycle is the non-convex ROA, as shown in the Appendix. According to the algorithm described in Section 3, we learn the two nonlinear dynamical equations with 100 hidden neurons. To ensure an accurate model, we use 9 million data points sampled on $(-1.5, 1.5) \times (-1.5, 1.5)$, and the learning rate of the training process varies from 0.1 to 1e-5 to

acquire a small enough $\alpha$. With the learned dynamics, we aim to find a valid Lyapunov function over the domain $\|x\|_2 < 1.2$, and the obtained neural Lyapunov function is shown in Fig. 2a. The corresponding ROA estimate can be found in Fig. 2b, as the black ellipse. For comparison, the ROAs found by the neural Lyapunov function and the classical LQR techniques in [22] using actual dynamics are also plotted, as the blue and magenta ellipses respectively. The phase portrait of the system is given as the grey curves with small arrows. It is obvious that the neural Lyapunov function after tuning obtains a larger ROA. We also have a comparable verified ROA for the system with the one obtained with actual dynamics based on the same neural Lyapunov approach, both larger than the LQR case. The values of the parameters can be found in Table 1. The dynamics, the neural Lyapunov function, and the nonlinear controller for all experiments are described in detail in the Appendix.

Table 1: Parameters in Van der Pol Oscillator case

| $K_f$ | $K_\phi$ | $\delta$ | $\alpha$ | $\|\frac{\partial V}{\partial x}\|$ | $\beta$ | $\varepsilon$ |
|-------|----------|----------|----------|--------------------------------------|---------|---------------|
| 3.4599 | 5.197 | 5e-4 | 8.5e-3 | 1.249 | 0.02 | 0.2 |

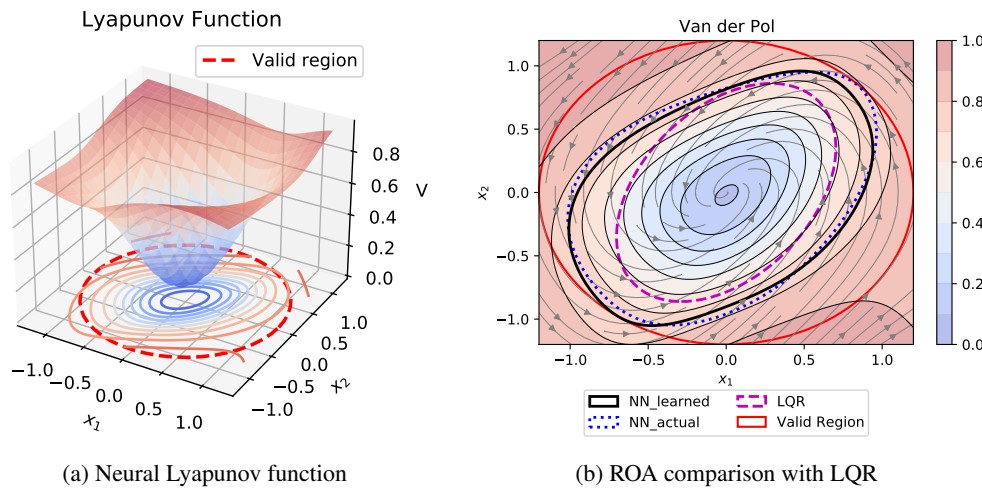

(a) Neural Lyapunov function      (b) ROA comparison with LQR

Figure 2: Neural Lyapunov function and the corresponding estimated ROA for Van der Pol oscillator.

## 5.2 Unicycle path following

The path following task is a typical stabilization problem for a nonlinear system. Here, we consider path tracking control using a kinematic unicycle with error dynamics [4]. With the learned dynamics $\phi$, a neural Lyapunov function can be learned on the valid region $\|x\|_2 \leq 0.8$, and the neural controller is set as $u = 5 \tanh(kx + b)$. The ROA comparison with the LQR method can be found in Fig. 3a. Apparently, the neural network method yields a larger estimated ROA, compared to the classical LQR approach in which the level set is determined by considering some relaxation of the largest reasonable range of linearization for practical systems under a small angles assumption ($< \frac{\pi}{9}$), given the fact that the actual dynamics is unknown.

The parameters for this valid neural Lyapunov function and the learned dynamics in this case are listed in Table 2. The Lipschitz constant $K_f$ is computed by using the bound $\|J_f\|_2 \leq \sqrt{m}\|J_f\|_\infty$, where $J_f$ is the Jacobian matrix of $f$ and $m$ is the number of rows of $J_f$. Note that $\alpha$ can be the maximum of the 2-norm loss over a test dataset as stated in Section 3.2, since what we need here is the discrepancies between the actual value and the approximated value of some known samples. In this regard, we can train FNN with fewer data samples, which is more computationally efficient. Then on top of that, a much larger dataset uniformly sampled over state and input spaces is used to calculate $\alpha$, which contributes to a smaller $\delta$. We implement the same approach for the inverted pendulum case as well.

Table 2: Parameters in unicycle path following case

| $K_f$ | $K_\phi$ | $\delta$ | $\alpha$ | $\|\frac{\partial V}{\partial x}\|$ | $\beta$ | $\varepsilon$ |
|---|---|---|---|---|---|---|
| < 45 | 108 | 1e-4 | 7e-3 | 4.43 | 0.1 | 0.1 |

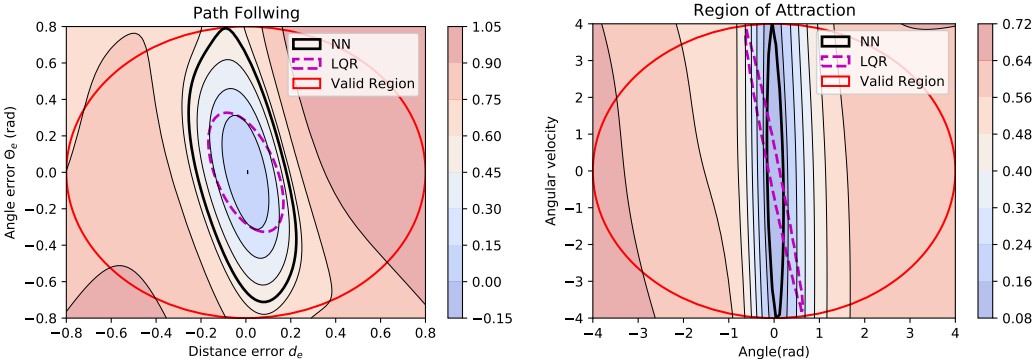

(a) ROA comparison for path following      (b) ROA comparison for inverted pendulum

Figure 3: Comparison of obtained ROAs for path following and inverted pendulum.

### 5.3 Inverted pendulum.

The inverted pendulum is another well-known nonlinear control problem. This system has two state variables $\theta$, $\dot{\theta}$ and one control input $u$. Here, $\theta$ and $\dot{\theta}$ respectively represent the angular position from the inverted position and angular velocity. The valid domain is $\|x\|_2 \leq 4$. The similar ROA comparison is shown in Fig. 3b, where the LQR approach uses the same function as given in [5]. The details regarding the bounds and $\beta$ are given in the Appendix.

## 6 Conclusion

This paper explores the ability of neural networks to approximate an unknown nonlinear system with a sufficient precision and find a neural Lyapunov function as well as a feedback controller to stabilize the unknown dynamics. Provable guarantees are also provided with the help of bounds on the approximation error and partial derivatives of the Lyapunov function. Moreover, experimental results show that the proposed approach outperforms the existing classical approach given part of the dynamics is known. The algorithm is only tested on low-dimensional systems. How to address this issue for high-dimensional systems is an interesting future research direction.

## Acknowledgements

This work was funded in part by the Natural Sciences and Engineering Research Council of Canada (NSERC), the Canada Research Chairs (CRC) program, and the Ontario Early Researcher Awards (ERA) program.

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
