# OpenReview forum: "Neural Lyapunov Control of Unknown Nonlinear Systems with Stability Guarantees"
_NeurIPS.cc/2022/Conference — NeurIPS 2022 Accept_

### Official Review · Reviewer_KQuS · 2022-06-23

**Rating:** 7
**Confidence:** 5
**Soundness:** 3 good
**Presentation:** 3 good
**Contribution:** 2 fair

**Summary:**

The paper addresses the challenging task of guaranteeing asymptotic stability of an unknown nonlinear system by simultaneously learning the unknown dynamics (via a neural network) and a valid Lyapunov function for the learned dynamics. They provide an extension of the method proposed by Chang et. al (ref. [6]) to unknown nonlinear systems.

The problem is fundamental in the sense that it bridges the gap between theoretical control theory and data-driven control by being able to guarantee strong stability properties for unknown and general nonlinear systems.


**Questions:**

I have the following issues with your approach

1) How do guarantee the existence of a feedback nonlinear controller that would steer the unknown nonlinear system to the origin? For instance, the recent work by Zinage et al. titled “Neural Koopman Lyapunov control” simultaneously learns a Koopman-based bilinear model and a valid Control Lyapunov Function (CLF) for the unknown nonlinear system and consequently guarantees the existence and computes a feedback controller by using the universal Sontag’s formula (provides necessary and sufficient conditions for asymptotic stability for controlled nonlinear systems) which is well studied in the control theory literature.

2) How do you ensure that your designed nonlinear controller would ensure forward invariance for the unknown nonlinear system? In other words, how do you ensure your controller will keep the trajectories within the ROA?

3) The SMT solver is not scalable and hence the computational time needed to compute a valid Lyapunov function increases exponentially as the dimension of the state increases. Consequently, this would restrict your approach to toy examples and therefore the applicability of your approach to real-world robotics problems like humanoids, manipulators, etc. becomes nontrivial.

4) One of the Lyapunov conditions is that V(0)=0. This is a “hard constraint”. However, this condition is added as a soft constraint in your loss function. Hence after the learning process is over, V(0) might not be “exactly” zero which violates one of the Lyapunov conditions thereby not making it to learn a valid Lyapunov function.

5) How is the value of beta chosen? It is mentioned that beta is kept relatively high so as to account for the approximation errors between the unknown nonlinear system and the learned system. I believe that if that is the case it might be sometimes difficult for the SMT solver to satisfy one of the Lyapunov conditions involving beta. It would be beneficial to add a table describing the influence of beta on the computational time required by the SMT solver to generate counterexamples.

6) How do you guarantee the learning process finally converges i.e it is possible to reach a state where there are no more counterexamples that satisfy the falsification constraint?

7) The proof of Theorem 4 is basically from the logic presented in Section 3.2 (Eqns. 8,9 and 10). I would suggest that you add this theorem before Eqns. 8, 9, and 10 and remove it from Appendix (page 14)

8) For some nonlinear systems, there does not exist any linear feedback controller that could stabilize the nonlinear system (however a nonlinear feedback controller exists). In view of that, the controller design in your paper is restricted to only a linear feedback controller (Lines 12-18 of Algorithm 1) which might not steer the nonlinear system in some cases.

9) Please elaborate on how Theorem 6 helps in the proof.

10) I think some important references are missing which are relevant with your work. For instance, the papers titled
“ Neural Lyapunov redesign”
“A ‘universal’ construction of artstein’s theorem on nonlinear stabilization”
“ Neural Koopman Lyapunov Control”



**Limitations:**

I would suggest adding a section to discuss the limitations of this approach. For instance, based on the following question
1) Does the learning framework finally converges?
2) SMT solver scalability?
3) Existence of linear feedback controllers?
4) Forward invariance when a linear controller is used?

**Strengths And Weaknesses:**

The strength of the paper is as follows:
1) To the best knowledge of mine, this is the first paper that attempts to provide stability guarantees for unknown nonlinear systems

The weakness of the paper is as follows:
1) Since the SMT solver is not scalable, the approach is restricted to low-dimensional problems and might not be suitable for real-world robotics applications

2) The convergence of the algorithm depicted in Figure 1 is not clear.

More details about the approach are presented under "Questions"

---

> ### Author Response · Authors · 2022-08-02
> **Author Response to Reviewer KQuS (1/3)**
>
> We greatly appreciate your valuable and detailed comments. We have updated our '**Related work**' part and added a new subsection to the appendix based on your comments, highlighted in blue.
>
> Due to space constraints, we would like to address all your concerns with a couple of comments, which follow below.
>
> ### Weakness
> 1. It is worth mentioning that the scalability of SMT solvers is still one of the hot topics in the area of neural network verification. We have added one subsection, Appendix D: Limitations and future work, in the manuscript to state this limitation in detail. For now, this approach mainly suffers from learning a precise neural network approximation for high-dimensional systems, as stated in the appendix. But we have seen the feasibility of learning a six-dimensional quadrator in some literature, for instance, [[1]](1). Moreover, we have seen dReal works well for verifying a six-dimensional system in [[2]](2). Therefore, we believe, the verification with SMT solvers should be feasible for some normal real-world robotics applications, such as, quadrators or bicycle-like vehicles.
>
>
> 2. The algorithm is not guaranteed to find a Lyapunov function if one exists. To the best knowledge of the authors, there is no algorithm that offers such guarantees, including the original '*Neural Lyapunov Control*' paper [[2]](2) and the '*Neural Koopman Lypunov Control*' [[3]](3) paper. What the proposed algorithm guarantees is that, once a Lyapunov function is found and verified to satisfy the proposed conditions, then the original *unknown* system is guaranteed to satisfy the asymptotic stability condition. We are not aware of any approach that offers such guarantees and our result is the first to provide closed-loop stability guarantees for systems with unknown dynamics.
>
> ### Questions
> Thank you for taking the time to go through our paper and leaving many detailed questions.
>
> 1. We greatly appreciate this question and the paper you mentioned here. The idea of using Koopman observables to design a stabilizing controller and derive a control Lyapunov function is interesting. After mapping the discrete-time system into a linear space, it is potentially much easier to study the stability with Lyapunov theorems for the unknown nonlinear systems. We were aware of this paper and spent some time studying it when it was just published on arXiv at the beginning of 2022. We also tried to run the code to reproduce the results. However, we found two small but significant mistakes in the code: one is in the second cell where the lie derivative of $V$ should be with respect to all $x$, instead of $x_1$ only; the other is in the sixth cell *Learning and Falsification*, when calculating *Koopman based bilinear system of Variable dReal*, '+' sign here gives the concatenation of the two lists in Python, but apparently that should be a summation, where *numpy.sum* is recommended. After we fixed these two mistakes, the results obtained using the code posted on Github differ significantly from the plots shown in the paper. The controller found by the code leads to unstable trajectories. This is one of the reasons why we did not cite this paper in our original manuscript. Moreover, the control Lyapunov function in that paper is verified for the lifted Koopman observables, not the original states, which means, it cannot provide stability guarantee for the original unknown system, while our paper addresses this issue. For this reason, the result of this paper is not directly comparable to ours. But the idea of learning a stabilizing controller for the unknown system is great. \
> Regarding how to guarantee that the obtained feedback nonlinear controller can make the system asymptotically stable, we rely on the Lyapunov theorems. We verify that all Lyapunov conditions are met for the unknown systems, and then the control Lyapunov function can guarantee the system is asymptotically stable within the sub-level set of the Lyapunov function that is completely contained within the ROA. Surprisingly, this verification problem is nuanced and quite technical and is referred to in greater detail in our answer to your Question 4.
>
> [1] Bansal, Somil, et al. "Learning quadrotor dynamics using neural network for flight control." 2016 IEEE 55th Conference on Decision and Control (CDC). IEEE, 2016.
>
> [2] Chang, Ya-Chien, Nima Roohi, and Sicun Gao. "Neural lyapunov control." Advances in neural information processing systems 32 (2019).
>
> [3] Zinage, Vrushabh, and Efstathios Bakolas. "Neural Koopman Lyapunov Control." arXiv preprint arXiv:2201.05098 (2022).

---

> > ### Author Response · Authors · 2022-08-02
> > **Author Response to Reviewer KQuS (2/3)**
> >
> > ### Questions (cont.)
> > 2. The goal of the approach is to compute a control Lyapunov function. Once the Lyapunov function has been learned and verified by the SMT solver, we use the level sets of the Lyapunov function to certify that the region these level sets enclose is forward invariant. This is a standard result in stability theory, which is stated in the well-known *Nonlinear Control* book by Hassan K. Khalil [[4]](4). That is, all the trajectories that start within the ROA will remain in the ROA and go to 0 eventually, as described in Definition 2.
> >
> > 3. We believe that this question is similar to the weakness about scalability, which has been addressed above. We will greatly appreciate it if the reviewer could regard the explanation in the response to **Weakness** as a proper answer to this question as well.
> >
> > 4. We agree that the condition "V(0) = 0" is a hard constraint and has to be enforced for this to be considered a Lyapunov function. However, the Lyapunov conditions are not satisfiable in a sufficiently small neighborhood of the origin by the SMT solver, so the condition V(0) = 0 is not computationally enforceable. We just encourage the value at 0 to be as close to zero as possible after training as a small value at 0 typically results in a larger ROA. This issue is non-trivial and the lack of V(0) = 0 does not even imply that some ball around the origin is asymptotically stable. This problem was not addressed in the '*Neural Lyapunov Control*' [[2]](2) paper, which is reference [5] in our manuscript, but we discuss reasonable assumptions, Assumptions 3 \& 4, for the ball of the origin to be asymptotically stable in Theorems 3 and 4.
> >
> > 5. Since we are performing system identification, we also require rigorous error estimates so that the practical Lyapunov function (Lyapunov conditions verified outside some ball of the origin) outputted by the neural network can be shown to be the practical Lyapunov function of the actual dynamics. Given that neural networks can be computationally demanding, we need to balance the error between the value of beta and the Lipschitz constants of the actual dynamics and the neural network. With a larger value of $\beta$, the condition becomes more difficult for the SMT solver to satisfy and the training process takes more time. However, the learning and verification process is consistent with the one observed in [[2]](2), apart from the determination of the values of the parameters. We did not notice a significant increase in computational time (definitely not an exponential increase).
> >
> > 6. Computing a valid Lyapunov function is challenging and has been a well-studied topic for dynamical systems. We admit that our method is not complete, that is it does not guarantee that we can obtain a valid Lyapunov function after running the algorithm. The original paper [[2]](2) similarly suffers from this same issue and to the best of our knowledge, we are unaware of any papers in the literature that have addressed this issue. Finding a complete algorithm for computing Lyapunov functions would warrant its own paper and is a topic we will consider in the future.
> >
> > 7. Thank you for the insightful comment. When we were writing the paper, we considered putting Theorem 4 in Section 3.2, but later we found the precise statement heavily relies on Assumptions 3 \& 4, and it is, therefore, better to have a separate section (Section 4) to state all the theoretical results clearly and systematically. Moreover, in order to make our proof in the appendix easy to follow, we simply restate this theorem.
> >
> > 8. We appreciate your valuable insights on this, and we agree with your question. This is the reason why we designed a nonlinear controller to stabilize the nonlinear system with a saturation feature to simulate real motors in real-life applications, and consequently, the bounded input feature of the controller makes learning the unknown dynamics possible. The initial controller in this setting is the linear one given by LQR, however, the controller is updated in the training process in line 14 of the algorithm and is depicted in Figure 1. Since the controller is the output of a neural network, it is, therefore, a nonlinear function.
> >
> > 9. Theorem 6 is a supplementary result that holds for the case of radially unbounded Lyapunov functions. This result is not used in the proofs of the other theorems but says that we are able to recover the compact set $K$ as the level sets tend to infinity.
> >
> > [2] Chang, Ya-Chien, Nima Roohi, and Sicun Gao. "Neural lyapunov control." Advances in neural information processing systems 32 (2019).
> >
> > [4] Khalil, Hassan K. Nonlinear control. Vol. 406. New York: Pearson, 2015.

---

> > > ### Author Response · Authors · 2022-08-02
> > > **Author Response to Reviewer KQuS (3/3)**
> > >
> > > ### Questions (cont.)
> > >
> > > 10. Thank you for bringing these papers to our attention. All of them are good papers. Among them, '*Neural Lyapunov redesgin*' and '*Neural Koopman Lyapunov Control*' are strongly related to our work. We have cited them properly in the updated manuscript, while the paper titled '*A universal construction of Artstein’s theorem on nonlinear stabilization* by ED Sontag is mainly about how to design a control Lyapunov function for control-affine systems. The main focus of this paper is about general nonlinear systems, and the corresponding useful existing theorems for control Lyapunov functions of nonlinear systems are stated and well-explained in [[4]](4), which is cited as reference [21] in the paper.
> > >
> > > ### Limitations
> > > We genuinely thank the reviewer for this helpful comment. We have added the discussion on the limitations of this work to the supplementary material, as **Section 7.4 Appendix D: Limitations and future work**. Please check the updated manuscript for details.
> > >
> > > [4] Khalil, Hassan K. Nonlinear control. Vol. 406. New York: Pearson, 2015.

---

> > > > ### Comment · Reviewer_KQuS · 2022-08-03
> > > > **Response to authors comments**
> > > >
> > > > Thank you for your responses to my comments.
> > > >
> > > > 1) Regarding the code implementation of the paper titled “Neural Koopman Lyapunov Control”: If you read their paper carefully, you will realize that it is derivative of V with respect to z and not with x or x1. What they have done in their code is first they have taken the derivative of V wrt x1 and then the derivative of z wrt to x1. These two values are divided to obtain the value of dV/dz:=(dV/dx1)/(dz/dx1). So when you mention that they have taken derivative wrt x1 and not x. Both are actually not true. They have actually taken it wrt z. Also, the sixth cell Learning and Falsification, when calculating the Koopman-based bilinear system of Variable dReal, the '+' sign is the addition of the nominal and the drift dynamics of the bilinear model which is similar to that done in the paper titled “Neural Lyapunov Control”
> > > >
> > > > When you mention “Regarding how to guarantee that the obtained feedback nonlinear controller can make the system asymptotically stable, we rely on the Lyapunov theorems.”: When a nonlinear controller is involved, the notion of Control Lyapunov functions (CLF) comes into the picture and not Lyapunov Functions (and its theorems) which are both different in their definitions (for the definition of CLF's please refer to works by Sontag et al.). CLF provides necessary and sufficient conditions for guaranteeing asymptotically stable for controlled systems (Universal Sontag’s formula). From reading your paper,  I strongly believe you are computing a valid Lyapunov function and not a CLF. The change that has to made is $\nabla_f V(x)\dot{x}<0$ (Lyapunov function) to $\underset{u}{\inf}\quad \nabla_f V(x)\dot{x}<0$ (CLF) which differentiates it from the Lyapunov function. I suggest the authors to verify and add this change.
> > > >
> > > > Then you mention “We verify that all Lyapunov conditions are met for the unknown systems, and then the control Lyapunov function….”: Here you using CLF but there is no mention of CLF in your paper.
> > > >
> > > > 2) You mentioned, “The goal of the approach is to compute a control Lyapunov function. Once the Lyapunov function has been learned”. First, you mention that you are computing CLF and then you come back to the Lyapunov function (both are different in the control theory community). Please strive to make this distinction clear in your manuscript and suggest not using CLF and LF interchangeably. And yes existence of CLF (and not Lyapunov function for controlled systems) guarantees that the trajectories are within ROA. I think the result in the book by Khalil talks about uncontrolled nonlinear systems. Please verify.
> > > >
> > > >  Points 3,4,5 are convincing.
> > > >
> > > > Regarding point 6, I suggest you add that in one of the remarks of your manuscript.
> > > >
> > > > Points 7,8, 9, and 10 are convincing enough.
> > > >
> > > > Once changes to points 1, 2, and 6 are made as described above, I would like to accept this manuscript.

---

> > > > > ### Author Response · Authors · 2022-08-05
> > > > > **Response to Reviewer KQuS (1/2)**
> > > > >
> > > > > Thanks for the reviewer's prompt and insightful comments. We would like to address the issues mentioned in points 1 and 2 as follows.
> > > > >
> > > > > 1. We apologize for not making this clear in our previous response. For the mistakes in the code we mentioned before:
> > > > >   + The first mistake:
> > > > >   > in the second cell where the lie derivative of $V$ should be with respect to all $x$, instead of $x_1$ only.
> > > > >   >
> > > > >   What we want to express is when we are calculating the partial derivative $\frac{\partial V}{\partial z}$, we need to use chain rule from calculus:  $\frac{\partial V}{\partial z} = \frac{\partial V}{\partial x} \frac{\partial x}{\partial z}$ which involves all coordinates of $x$. $\frac{\partial V}{\partial x_1} \frac{\partial x_1}{\partial z}$ does not return the correct results. Here is an example: \
> > > > >     Let's assume $z$ and $x$ have the same dimension so $\phi(x)$ can indeed be invertible. We assume $V(x)=x_1^2+x_2^2, z_1=x_1+x_2$, and $z_2 = x_1-x_2$, so that $\phi$ is a linear transformation. Then, substituting gives $V = \frac{(z_1+z_2)^2}{4} + \frac{(z_1-z_2)^2}{4} = \frac{z_1^2}{2} + \frac{z_2^2}{2}$. Hence, $\frac{\partial V}{\partial z} = [z_1,  z_2]$, while $\frac{\partial V}{\partial x_1} = 2x_1$, and $\frac{\partial z}{\partial x_1} = [1; 1]$. \
> > > > > On the other hand, $\frac{\partial V}{\partial x} = [2x_1,  2x_2]$ and $\frac{\partial z}{\partial x}$ = [1,   1;1,  -1], then we have $\frac{\partial V}{\partial x} \frac{\partial z}{\partial x}^{-1}$ = $[2x_1, 2x_2]\frac{1}{2}$$[1,   1;1,  -1]$ = $[x_1+x_2,  x_1-x_2]$ = $[z_1, z_2]$
> > > > >
> > > > >   + The second mistake:
> > > > >   > in the sixth cell *Learning and Falsification*, when calculating *Koopman based bilinear system of Variable dReal*, '+' sign here gives the concatenation of the two lists in Python, but apparently that should be a summation, where *numpy.sum* is recommended.
> > > > >   >
> > > > > According to the equation (7) in the paper, the Koopman Bilinear Form for the lifted state $\boldsymbol{z}$: $\boldsymbol{z_{k+1}}=\mathcal{K_d}\boldsymbol{z_k}+\sum_{i=1}^{m} B_{i}\boldsymbol{z_k}u_i$, where $\boldsymbol{z_{k+1}}$ should have the same dimension as $\boldsymbol{z_{k}}$ for the bilinear system. We believe that they try to use '+' operation to perform the element-wise addition of $\mathcal{K_d} \boldsymbol{z_k}$ and $\sum_{i=1}^{m}B_i \boldsymbol{z_k} u_{i}$ in the code.
> > > > >
> > > > > However, after downloading the code and running two examples, we found f_koop in the sixth cell returns a list whose length is twice the dimension of the lifted state space. Take the Van der Pol code as an example, the dimension of $\boldsymbol{z_k}$ is 5, while the dimension of the output of Koopman linear system, f_koop, is 10. This is because the format of f_koop and the two summands is **list**. And, a '+' operation on two lists in Python returns a concatenation of the two lists, as illustrated below.
> > > > > ```
> > > > > A = [1, 2]
> > > > > B = [3, 4]
> > > > > > A + B = [1, 2, 3, 4]
> > > > > C = numpy.sum([A,B],axis = 0)
> > > > > > C = array([4, 6])
> > > > > ```
> > > > > Therefore, the f_koop in the verification process in the loop is just the first 5 elements in the list, which is $\mathcal{K_d} \boldsymbol{z_k}$ only, no input involved, which is not correct. Moreover, we don't think the original 'Neural Lyapunov Control' suffers from this, as they do not have '+' operation on lists.
> > > > >
> > > > > 2. As the second half of point 1 is similar to point 2, we address them together here. We agree with the reviewer on this. We are actually learning Lyapunov functions for the controlled systems for a fixed $u = \kappa(x)$, and it is worth mentioning that once a Lyapunov function is found for the controlled system $f(x,u)$, it is a control Lyapunov function for $f$, according to the [book chapter](https://link.springer.com/content/pdf/10.1007/978-1-4471-0807-8_40.pdf) by Sontag on "Control-Lyapunov functions" [[1]](1). (Note that control Lyapunov functions are well defined for general nonlinear system $f(x,u)$, not only for control-affine systems $f(x)+g(x)u$.) Therefore,  it guarantees the asymptotic stability of the unknown nonlinear system. This is why, in the previous response, when we referred to: using the 'control Lyapunov function' what we mean is exactly the obtained Lyapunov function for the given controller. Hence, we can guarantee that the obtained nonlinear controller makes the system asymptotically stable. For these reasons, we elect to not use the notation of control Lyapunov function for our Lyapunov function $V$ in the main body of our manuscript. Moveover, it is worth noting that we cannot implement Sontag's formula on the nonlinear system of general format $f(x,u)$, but if the system is a control-affine system, a nonlinear controller can be obtained with Sontag's formula, as shown in the 'Neural Koopman Lyapunov Control' paper.
> > > > >
> > > > > [1] Sontag, Eduardo D. "Control-lyapunov functions." Open problems in mathematical systems and control theory. Springer, London, 1999. 211-216.

---

> > > > > > ### Author Response · Authors · 2022-08-05
> > > > > > **Response to Reviewer KQuS (2/2)**
> > > > > >
> > > > > > 6. Thank you again for this insightful comment. We have added this paragraph to the limitations and future work section of this work, as point 3 of Section 7.4 in the updated appendix.
> > > > > > > Computing a valid Lyapunov function is challenging and has been a well-studied topic for dynamical systems. We admit that our method is not complete, that is, it does not guarantee that we can obtain a valid Lyapunov function after running the algorithm. The original paper of this framework [2] similarly suffers from this same issue and to the best of our knowledge, we are unaware of any papers in the literature that have addressed this issue.  Finding a complete algorithm for computing Lyapunov functions is an interesting topic for future research.
> > > > > > >

---

> > > > > > ### Comment · Reviewer_KQuS · 2022-08-05
> > > > > > **Response to authors comments**
> > > > > >
> > > > > > Thanks for your response to my comments. All of my comments have been answered and I would be changing my decision to accept.

---

> > > > > > > ### Author Response · Authors · 2022-08-07
> > > > > > > **To Reviewer KQuS**
> > > > > > >
> > > > > > > Thank you again for your insightful comments and engaging discussions. We are delighted to know that you are willing to update your score.

---

> > > > > > > > ### Author Response · Authors · 2022-08-09
> > > > > > > > **Thanks for your response and insightful comments (to Reviewer KQuS)**
> > > > > > > >
> > > > > > > > Dear Reviewer KQuS,
> > > > > > > >
> > > > > > > > Since the author/reviewer discussion period will end tomorrow, if there is anything else you would like us to clarify, please let us know before then. Also, we would greatly appreciate it if you could adjust your score so that we know your stance on the paper before the discussion period ends.

---

### Official Review · Reviewer_psi4 · 2022-06-30

**Rating:** 6
**Confidence:** 3
**Soundness:** 3 good
**Presentation:** 2 fair
**Contribution:** 3 good

**Summary:**

This paper proposed a framework to learn the non-linear system and control it with a neural controller and certified Lyapunov function. The system asymtotic stability can be proved by the negative Lie derivative of Lyapunov function. The satisfiability of Lyapunov function is achieved by running the SMT solver and learning framework iteratively. The effectiveness is demonstrated by 3 illustrative experiments.

**Questions:**

1. In Assumption 1 and line 166, the distance of (x,u1) and (y,u2) is a bit ambiguous with undefined “-” operator. It’s better to denote (x,y) as the concatenated vector for clarification.
2. Can you provide the proof for theorem 5 from your reference [29]?
3. Theorem 2 claims the existence of 1-hidden-layer neural network $\phi$ to approximate the unknown dynamics, based on universal approximation theory. However, this only proves the existence but not the actual learn  $\phi$ would satisfy such approximation accuracy. Similar for theorem 5. For instance, in line 503 of appendix, I’m wondering how to prove the actual $V_{\phi}$ satisfy the Lyapunov condition given theorem 5 only claims the existence?
4. I’m curious how to approximate M in line 167. If upper bound of $\lVert \frac{\partial V}{\partial x}\rVert $is approximated by sampling then this is not a guarantee. Same question applied to other Lipschitz constants.

**Limitations:**

This paper does not have potential negative societal impact.

**Strengths And Weaknesses:**

Strengths

1. The paper structure is easy to follow. The experiments are well organized and helpful for theorem illustration.
2. Under the framework of [1], The paper provides the proof of Lyapunov function existence and stability guarantee by quantifying the errors among unsampled states and nearest training samples.

[1] Ya-Chien Chang, Nima Roohi, and Sicun Gao. Neural Lyapunov Control. In arXiv:2005.00611 [cs, eess, stat], December 2020. arXiv: 2005.00611.

Weaknesses:

1. With many of the methods inherited from previous [1], other than the theoretical improvement, I suggest the authors to include the methodology difference of this paper and [1], along with the experimental comparison.
2. The formation of the problem(low dim system) and neural network structure are relatively simple. It would be interesting to see how the method applies in more complex system.
3. The major improvement of this paper is the theoretical proof of stability over the previous work. However, I’m confused by the derivation logic. See my question 3 and 4.

---

> ### Author Response · Authors · 2022-08-02
> **Author Response to Reviewer psi4**
>
> Thank you for your valuable feedback. We would like to address all mentioned concerns as follows.
>
> ### Weakness
> Thanks for the insightful comments about the weakness of this paper.
>
> 1. Addressing the difference between the '*Neural Lyapunov Control*' paper [[1]](1) and ours:
>
> Since this work is based on [[1]](1), the contributions as stated in the Introduction (Section 1) can be regarded as the main differences compared to the previous paper. We would like to restate the main differences clearly as follows:
>
>  + In the previous paper, they assume the nonlinear systems are known, while we assume the right-hand sides are all unknown. This renders the stability guarantee through the Lyapunov function challenging to verify directly. In our approach, we first learn the dynamics and then pose stricter Lyapunov conditions (involving the $\beta$ constant) to guarantee that the learned Lyapunov function for the learned dynamics can then provide stability guarantees to the original unknown dynamics by employing Lipschitz constants.
>  + Instead of a simple linear controller, we have a bounded nonlinear controller to improve its closed-loop performance. Moreover, this enables the learning of unknown dynamics with the bounded input feature as shown in Section 3.1.
>  + We provide rigorous proofs for all learning processes (learning the dynamics and the existence of the Lyapunov function) and stability guarantees for the unknown systems, which is not stated explicitly in the previous work, especially for one of the Lyapunov conditions $V(0) = 0$. Omitting this condition is quite delicate and we fix this by introducing reasonable assumptions, Assumptions 3 \& 4, in Section 4.
>
> 2. We agree that this is one of the main limitations of this paper. We will try to implement this algorithm on high-dimensional systems for both simulations and real experiments in future work. Additionally, we have added a new subsection to the supplementary material, Section 7.4, to clarify the limitations of this work more clearly.
>
> 3. This comment will be addressed in the Questions section.
>
> ### Questions
> 1. In that line we are looking at the euclidean distance between two vectors $(x,u)$ where the first state lies in the state space and the second state lies in the input. The ''-'' sign represents a minus and not a concatenation.
>
> 2. Since this is an important result, including the proof for completeness would enhance the paper. However, we feel that since the proof would be too long as we have to include several key lemmas, it would take away from our other contributions in the supplementary material so we choose to omit it and refer the reader to the excellent review paper.
>
> 3. Theorem 2 asserts that the problem is a well-defined one since we show that we are able to get arbitrarily close to the Lipschitz constant of the vector field. However, for our methodology to perform well we do not require an arbitrarily close Lipschitz constant. In the experiments part, we observe that the Lipschitz constant of the approximated dynamics, $K_\phi$ is not very close to $K_f$, however, they are close enough for the stability analysis to hold.
>
> 4. We use the SMT solver to verify M as an upper bound by gradually increasing its value. We calculate the Lipschitz constant $K_f$ algebraically. In the case of the Van Der Pol oscillator, the expression is simple. For the other two cases, we upper bound it by using the matrix-infinity norm as in Line 623 (updated version). We then calculate $K_\phi$ by using the LipSDP-network algorithm as in Line 178 (updated version). While calculating $K_f$ would invite criticism since it is assumed to be unknown, this method is reasonable under our controlled study since we always observed that the Lipschitz constant of the neural network, $K_\phi$ is larger. This is due to the conservative estimation of the Lipschitz constants calculated through the LipSDP-network algorithm.
>
>
> [1] Chang, Ya-Chien, Nima Roohi, and Sicun Gao. "Neural lyapunov control." Advances in neural information processing systems 32 (2019).

---

> > ### Comment · Reviewer_psi4 · 2022-08-05
> > **response to author rebuttal**
> >
> > Thank you for the reply.
> >
> > Most of my concerns are addressed. As for question 3, I intended to ask how to prove the convergence of searching such Lyapunov function, your answer to other reviewer partially clarify it. Despite there is not a guarantee, the proof for existence and verification are still of significance to the community. I will move my position to accept.
> >
> > Some minor points:
> >
> > Regarding question 1, I suggest explicitly denoting $(x,u)$ as the concatenation.
> >
> > Regarding question 2, can you point out which Theorem from the long review paper did you refer to? I'm interested in the proof though.

---

> > > ### Author Response · Authors · 2022-08-08
> > > **Response to Reviewer psi4**
> > >
> > > We thank the reviewer genuinely for the reply and for changing the score. In response to your points:
> > > 1. Thanks again for this comment. We have denoted $(x,u)$ as the concatenation in the updated manuscript, highlighted in blue in Line 108.
> > > 2. Regarding the proof, we are referring to Theorem 4.1 of the long paper [[1]](1). The statement of Theorem 4.1 is more general than Theorem 5 in our paper. In the review paper, they consider multi-index notation that is typical for PDEs. Namely, $\mathbb{Z_+^n}$ is the lattice of nonnegative multi-integers, that is, $\textbf{m} = (m_1,...,m_n) \in \mathbb{Z_+^n}$ if each $m_i \in \mathbb{Z}$. On the lattice there is the standard partial ordering, namely $\mathbf{m}^{1} \leq \mathbf{m}^{2}$ if $m_{i}^{1} \leq m_{i}^{2}$, $i=1, \ldots, n$. For notation purposes, set $|\mathbf{m}|=m_{1}+\cdots+m_{n}, \;\mathbf{x}^{\mathbf{m}}=x_{1}^{m_{1}} \cdots x_{n}^{m_{n}}$. We introduce this notation so that we can take arbitrary partial derivatives: \begin{equation}
> > > D^{\mathbf{m}}=\frac{\partial^{|\mathbf{m}|}}{\partial x_{1}^{m_{1}} \cdots \partial x_{n}^{m_{n}}}
> > > \end{equation}
> > > and consider a more general class of function spaces where we can demand greater regularity/smoothness. We say $f \in C^{\mathbf{m}}\left(\mathbb{R}^{n}\right)$ if $D^{\mathbf{k}} f \in C\left(\mathbb{R}^{n}\right)$ for all $\mathbf{k} \leq \mathbf{m}, \mathbf{k} \in \mathbb{Z_+^n}$, meaning that we can take partial derivatives of all multi-index values less than $\textbf{m}$. The function space we take is
> > > \begin{equation}
> > > C^{\mathbf{m}^{1}, \ldots, \mathbf{m}^{s}}\left(\mathbb{R}^{n}\right)=\bigcap_{j=1}^{s} C^{\mathbf{m}^{j}}\left(\mathbb{R}^{n}\right).
> > > \end{equation}
> > > If $f \in C^{\mathbf{m}^{1}, \ldots, \mathbf{m}^{s}}\left(\mathbb{R}^{n}\right)$, then that means that $D^{\mathbf{k}} f \in C\left(\mathbb{R}^{n}\right)$ for all $\mathbf{k} \leq \mathbf{m}^i$ for some $i= 1,\cdots,s$. In Theorem 5 of our paper, we take $s = n$ and $\textbf{m}^i = (0,\cdots,1,\cdots)$, where the 1 is in the $i$th position. In this case, $D^{\textbf{m}^i} = \frac{\partial}{\partial x_i}$.  The statement of Theorem 4.1 is: \
> > > Let $\mathbf{m}^{i} \in \mathbb{Z_+^n}, i=1, \ldots, s$, and set
> > > $m=\max$ \{ $|\mathbf{m}^{i}|: i=1, \ldots, s$ \}.
> > > Assume $\sigma \in C^{m}(\mathbb{R})$ and $\sigma$ is not a polynomial. Then $\mathcal{M}(\sigma)$ is dense in $C^{\mathbf{m}^{1}, \ldots, \mathbf{m}^{s}}\left(\mathbb{R}^{n}\right)$. \
> > > In their notation, $\mathcal{M}(\sigma)$ is the space of one layer neural networks. We say that $\mathcal{M}(\sigma)$ is dense in $C^{\mathbf{m}^{1}, \ldots, \mathbf{m}^{s}}\left(\mathbb{R}^{n}\right)$ if, for any $f \in C^{\mathbf{m}^{1}, \ldots, \mathbf{m}^{s}}\left(\mathbb{R}^{n}\right)$, any compact $K$ of $\mathbb{R}^{n}$, and any $\varepsilon>0$, there exists a one layer neural network $\mathcal{M}(\sigma)$ satisfying
> > > \begin{equation}
> > > \max_{\mathbf{x} \in K}\left|D^{\mathbf{k}} f(\mathbf{x})-D^{\mathbf{k}} g(\mathbf{x})\right|<\varepsilon,
> > > \end{equation}
> > > for all $\mathbf{k} \in \mathbb{Z_+^n}$ for which $\mathbf{k} \leq \mathbf{m}^{i}$ for some $i$. Therefore, to be compared with our Theorem 5, this means that when $s = n$ and $\textbf{m}^i = (0,\cdots,1,\cdots)$, we are able to approximate a sufficiently smooth function and its first order partial derivatives simultaneously since this approximation has to hold for all smaller multi-indexes at the same time (assuming that the activation is also sufficiently smooth, which is true for us as we use tanh activation functions).\
> > > The proof of this statement is detailed and will require knowledge of the standard universal approximation theorem, a read of Theorem 3.2 and a read of the statement and proofs of Proposition 3.3. In the proof of Theorem 4.1, we think there is a typo at the bottom of page 164 and we beleive that it should instead read as:
> > > \begin{equation}
> > > \left\||f-g\right\||_{C^{m}[\alpha, \beta]}<\varepsilon
> > > \end{equation}
> > > Moreover, the author skips a piece of detailed analysis at the end of the proof, and if you are interested this corresponds to Section 3 of [[2]](2).
> > >
> > > [1]Pinkus, Allan. "Approximation theory of the MLP model in neural networks." Acta numerica 8 (1999): 143-195.
> > >
> > > [2] Li, Xin. "Simultaneous approximations of multivariate functions and their derivatives by neural networks with one hidden layer." Neurocomputing 12.4 (1996): 327-343.

---

### Official Review · Reviewer_Y9Y1 · 2022-07-11

**Rating:** 6
**Confidence:** 3
**Soundness:** 3 good
**Presentation:** 3 good
**Contribution:** 3 good

**Summary:**

This paper proposes a neural architecture that can (1) learn dynamics, (2) find a Lyapunov function and corresponding nonlinear stabilizer simulatenously. Using a satisfiability modulo theories (SMT) solver is also applied to verify the Lyapunov conditions of the candidate Lyapunov function. The authors claim that simultaneous learning of both the function and stabilizer helps to draw stability guarantees and overall better performance. This paper contains detailed theoretical guarantees and multiple experimental evidence, including comparison studies with existing methods.


**Questions:**

1. I am interested in using a solver in the algorithm. How did you choose an SMT solver for the experiments? Was there an extenstive ablation study for choosing an appropriate solver?


**Limitations:**

Even though most of the claims are sound, I also believe that the amount of experiments that are shown in this paper is limited, and the network sizes used in this work are relatively small. I am concerned about the general usability of this model for the complex nonlinear dynamical system, which is a core claim of this paper. The generality of the method needs to be validated with bigger data and larger neural networks.


**Strengths And Weaknesses:**

This is overall a solid paper that retains theoretical and empirical novelties for challenging nonlinear control problems. This paper provides detailed theoretical guarantees and various experimental results. Meanwhile, I have also noted some concerns about the proposed methodology. I hope to see a response to these points during the rebuttal.


### Strengths

* This paper is quite well written. The authors put a lot of effort into providing relevant information in the Introduction and Preliminaries sections. The algorithm and learning schemes were easy to follow.
* As far as I know, the proposed neural network architecture and algorithm are novel. These can contribute to finding the Lyapunov function for learning real-time physical systems with unknown dynamics. The theoretical results presented in this paper are sound.
* This work presents trainable nonlinear controllers and Lyapunov function using neural networks. I think involving an SMT solver in the algorithm is a good approach.

### Weaknesses

* In the algorithm, I think the part of learning dynamics is not polished. The authors make assumptions that the dynamics are unknown. In this sense, oftentimes, it will be hard to collect trainable data, and it cannot be appropriate for a neural network to train dynamics by only using SGD. It seems that the overall performance of the algorithm is tightly coupled with the performance learned dynamics $f_\phi$. Although Theorem 2 ("an extension of the universal approximation theorem") is presented, I think there are some real challenges that dynamics learning struggles. I can imagine multiple realistic issues regarding model complexity and data, so the learning dynamics might fail.

* In the experiments, it would be better if there was a metric to compare ROA estimation performance between the proposed approach and LQR. I cannot grasp "how much" learning the Lyapunov function outperforms LQR.

---

> ### Author Response · Authors · 2022-08-02
> **Author Response to Reviewer Y9Y1**
>
> The authors genuinely appreciate your valuable comments. We have revised our paper based on your comments, with major changes highlighted in blue. We also would like to answer all your concerns directly here.
> ### Weakness
> Thanks for pointing out the limitations of this manuscript.
>    + We agree that one of the main weaknesses of this work is that we assume a large number of exact measurements are available for training which might not be possible in practice. We have added one subsection in the supplementary material to clearly state this limitation, and the submitted PDF file has been updated accordingly. As described in the appendix, we focus on showing the proposed approach works for learning and stabilizing the unknown nonlinear system in the case of exact measurements, and we plan to extend the results to the case of noisy measurements in the future.
>   +  Regarding the comparison between the proposed method and LQR, generally speaking, the *metric*, or we call it performance measure, should be the size (area) of the ROA, as described in [[1]](1), which is well depicted in the figures in the Experiments (Section 5). This is because we always want more states in the state space to be stabilized. With that said, in the authors' opinion, the ellipsoid plots in the paper are able to illustrate the comparisons well in this regard.
>
> ### Questions
>   1. The authors would like to recommend scholars who are interested in choosing a suitable SMT solver for their research to have a look at the results of the latest SMT solver competition. The previous results can be found by following the link: [SMT COMP](https://smt-comp.github.io/2021/results.html). However, choosing the best SMT solvers is problem-dependent, but the competition results can serve as a guide.
> For our specific problem,  dReal is the best fit given the fact that we have two neural networks together to verify, both of which have hyperbolic $tanh$ activation functions. We also explored two other state-of-the-art SMT solvers, Z3 and CVC5, which had excellent performances in the competition. But they could not handle this task very well, either taking too long or failing to return the results. Therefore, we use the same SMT solver, dReal, as in the previous work [[2]](2), reference [5] in the paper.
>
> ### Limitations
>   + We totally agree with the limitations mentioned here, and we greatly appreciate your valuable insights. As stated before, the aim of this paper is to show the effectiveness of the proposed algorithm on some classic nonlinear control problems given sufficiently many exact measurements. We will try to address these issues in our future work. Again, we have included the limitations of this work in **Section 7.4 Appendix D: Limitations and future work**.
>
> [1] Mehrjou, Arash, Mohammad Ghavamzadeh, and Bernhard Schölkopf. "Neural lyapunov redesign." arXiv preprint arXiv:2006.03947 (2020).
>
> [2] Chang, Ya-Chien, Nima Roohi, and Sicun Gao. "Neural lyapunov control." Advances in neural information processing systems 32 (2019).

---

### Meta-Review · Area_Chair_TwdZ · 2022-08-25

**Recommendation:** Accept
**Confidence:** Certain

**Metareview:**

The problem statement and the technical approach are interesting. Some concerns about the scalability of the approach remain; however, given the novelty of the approach, I am recommending that the paper be accepted in spite of these concerns. Please make sure to incorporate the reviewers' feedback into the final version.

**Award:**

No

---

### Decision · Program_Chairs · 2022-09-14

Accept